# Constraint and divergence of global gene expression in the mammalian embryo

**Noah Spies[1,2†], Cheryl L Smith[1,2†], Jesse M Rodriguez[3,4‡], Julie C Baker[2], Serafim Batzoglou[3], Arend Sidow[1,2*]**

[1]Department of Pathology, Stanford University School of Medicine, Stanford, United States; [2]Department of Genetics, Stanford University School of Medicine, Stanford, United States; [3]Department of Computer Science, Stanford University, Stanford, United States; [4]Biomedical Informatics Program, Stanford University School of Medicine, Stanford, United States

**Abstract** The effects of genetic variation on gene regulation in the developing mammalian embryo remain largely unexplored. To globally quantify these effects, we crossed two divergent mouse strains and asked how genotype of the mother or of the embryo drives gene expression phenotype genomewide. Embryonic expression of 331 genes depends on the genotype of the mother. Embryonic genotype controls allele-specific expression of 1594 genes and a highly overlapping set of cis-expression quantitative trait loci (eQTL). A marked paucity of trans-eQTL suggests that the widespread expression differences do not propagate through the embryonic gene regulatory network. The cis-eQTL genes exhibit lower-than-average evolutionary conservation and are depleted for developmental regulators, consistent with purifying selection acting on expression phenotype of pattern formation genes. The widespread effect of maternal and embryonic genotype in conjunction with the purifying selection we uncovered suggests that embryogenesis is an important and understudied reservoir of phenotypic variation.

*For correspondence: arend@ stanford.edu

†These authors contributed equally to this work

Present address: ‡Google Inc., Mountain View, United States

**Competing interests:** The authors declare that no competing interests exist.

**Reviewing editor**: Anne C Ferguson-Smith, University of Cambridge, United Kingdom

## Introduction

Multicellular animals develop via complex pattern formation processes that unfold during embryogenesis, with gene regulation playing a central role. Surprisingly, the impact of genetic variation on gene regulation and the relative importance of the genetic mechanisms underlying regulatory variation in mammalian embryogenesis remain undefined. There are four broad categories of genetic mechanisms of gene expression regulation: imprinting, parental effects, cis-regulation, and trans-regulation. All of these are potentially subject to natural selection and provide genetic windows into the forces that shape gene regulatory evolution. Here, we quantify their impact and relevance in mammalian embryogenesis by asking which of these genetic mechanisms reveal evolutionary divergence or constraint since the last common ancestor of two distantly related mouse strains, C57BL/6 (B6) and Cast/Ei (Cast).

Genomic imprinting is a well-studied form of parent-of-origin-dependent gene expression that produces strongly biased allelic expression patterns in offspring according to the parental source of the inherited allele. The progeny's imprinted gene expression patterns result from epigenetic modifications put in place in the previous generation and reflect the particular genetic architecture at these specialized loci. A recent genome-wide survey estimated that fewer than 200 genes are imprinted in mice and humans (*Babak et al., 2008*). Imprinting is thought to play a critical role in early embryonic and placental development in mammals and has been shown to influence post-natal health and behavior [Reviewed in *Peters (2014)*].

**eLife digest** The way that the embryo of a mammal, such as a mouse or a human, develops from a fertilized egg is a complicated process that relies on controlling: which genes are active; when these genes activate; and for how long they are active. In broad terms, there are four ways that this control can be achieved:

First, inside the sperm or egg, genes can be marked with small chemical tags that flag these genes to be activated (or remain inactive) after fertilization, depending on whether the modification was made by the father (in the sperm) or the mother (in the egg); this process is known as 'imprinting'. Second, the mother can alter the gene activity in her offspring via the placenta; this process is known as 'maternal effect'. Third, instructions encoded within the embryo's DNA can directly control if, and when, a nearby gene becomes activated; this is known as 'cis-regulation'. Finally, similar instructions can also control genes that are situated elsewhere in the embryo's DNA through indirect mechanisms; this is known as 'trans-regulation'.

Now, Spies, Smith et al. have investigated these four processes in the offspring of two different strains of mice, one originally from Europe and the other from Southeast Asia. The two strains were crossbred and the resulting embryos were analyzed to see which of the four processes affected gene activity. This analysis revealed 31 imprinted genes and 331 genes that exhibited a maternal effect—which sometimes changed gene activity by as much as 52%. Spies, Smith et al. also found over a thousand DNA instructions in the embryo's DNA that could directly influence the activity of nearby genes, but fewer instructions that could indirectly control genes that were further away.

These results suggest that direct control of genes, which affects only the genes closest to the DNA instruction, can vary a lot between individual embryos of the same species. However, indirect control of embryonically active genes, which affects many genes across the genome at the same time, appears much more tightly constrained by evolutionary forces. Which genes in the mother are responsible for the molecular signals that drive the maternal effect is an important question for future work, with implications for the genetic basis of embryonic development and disease.

---

Related to but distinct from imprinting is the concept of parental effects. Parental effects produce phenotypic variation that depends, directly or indirectly, on the genotype of either the mother or father, rather than that of the offspring. Therefore, the resulting phenotypic patterns lag a generation behind the genetic transmission of the causal variants. The most well-studied parental genetic effects are caused by deposition of maternal transcripts into the egg prior to fertilization, resulting in differences in early embryonic development depending on the genotype of the mother. Certain genes have also been shown to respond to maternal influence after birth through genetically defined maternal behaviors (*Weaver et al., 2004*). Because of the small contribution, through the sperm, of the paternal transcriptome to the fertilized zygote, and because of the stronger maternal contribution to child rearing in most model organisms, parental effects are typically thought of as synonymous with maternal effects, although true paternal effects are known to exist (*Rando, 2012*).

Maternal effects have been shown to be important during embryonic development, leading to differences in the birth weight of mice depending on the genotype of the mother (*Cowley et al., 1989*; *Wolf et al., 2011*). However, neither the causal molecular mechanism for these effects in the mother nor the responding genes in the embryo have been identified and few studies to date have discriminated between prenatal and postnatal effects. Maternal effects likely contribute to the heritability of complex traits but have been confounded with family structure in mouse as well as human studies (*Mott et al., 2014*).

Genetic variants affecting gene expression are classified into cis or trans, depending on whether they act on the gene copy on the same chromatid (cis) or on both chromatids equally (trans). Regulatory variants near their target genes usually act in cis, by modifying the activity of a promoter or enhancer, but very occasionally act in trans if they affect a gene product that regulates a nearby locus. One approach to quantifying cis and trans regulation in model organisms involves assaying gene expression in two parental lines compared to allele-specific expression (ASE) in the hybrids (*Wittkopp et al., 2004*) and asking whether alleles are regulated differently in the parentals compared to the hybrids. In adult flies and mice, these analyses have demonstrated that cis and trans regulatory

variants frequently target the same genes, often with opposite, compensatory effects on the target's expression (*McManus et al., 2010*; *Goncalves et al., 2012*). Recent genome-wide studies of ASE patterns across many individuals have demonstrated the ability of high-throughput sequencing methods to directly identify genes with cis-regulatory variants (*Montgomery et al., 2010*; *Pickrell et al., 2010*).

Another method to quantify cis- and trans-regulatory variation is analysis of expression quantitative trait loci (eQTL) (*Schadt et al., 2003*). eQTL are identified as genetic loci whose genotypes correlate with gene expression changes across a number of genetically heterogeneous individuals. Genome-wide eQTL studies have demonstrated that the strongest genetic variation in expression regulation generally occurs in variants located proximally to their target genes, presumably acting in cis. While typically fewer in number, trans-regulatory variants affect both alleles of each target gene and can affect expression of many genes simultaneously, resulting in a large aggregate effect.

ASE and eQTL approaches have been applied widely in adult tissues in humans and model organisms. However, these approaches have not yet been applied to understand the global relationship between genetic variation and gene expression during mammalian embryogenesis, and while imprinting has been studied extensively, the contribution of maternal effects on mammalian embryonic gene expression has never been addressed. For several reasons, it is likely that patterns of regulatory divergence differ between embryos and adults. First, eQTL studies have demonstrated dynamic trans-regulatory variation during *Caenorhabditis elegans* larval development (*Francesconi and Lehner, 2014*). Second, recent studies in flies and fish have demonstrated higher expression of more ancient conserved genes as well as more conserved gene expression patterns during embryogenesis compared to the adult (*Domazet-Lošo and Tautz, 2010*; *Kalinka et al., 2010*). This suggests that the number of cis- as well as trans-eQTL should be lower in embryos compared to adult.

Here, we quantify evolutionary gene regulatory divergence between B6 and Cast during embryogenesis, addressing each of the four regulatory architectures. We focus on embryonic day 11.5 (E11.5), which is equivalent to roughly 40 days of gestation in human. It is a time characterized by organogenesis, rapid growth of the brain and limbs, increasingly complex and localized pattern formation, and consequently substantial constraints on allowable phenotypic variation. An ASE/eQTL hybrid approach facilitates quantifying the different types of regulation and their effects on expression variation, revealing unique and important features of gene regulatory architecture during mammalian embryonic development.

## Results

For this study, we chose two inbred strains derived from geographically separate subspecies of *Mus musculus*: C57BL/6J (B6), the classic inbred mouse reference strain, which was derived from European mice, and CAST/EiJ (Cast), which was derived from the Southeast Asian *Mus musculus castaneus*. Inbreeding caused random subsets of alleles segregating in their ancestral populations to become fixed, and as a consequence the two strains exhibit one single nucleotide polymorphism (SNP) difference every 120 bp on average (*Keane et al., 2011*).

We collected 16 E11.5 F1 embryos obtained from reciprocal crosses between B6 and Cast (*Figure 1*). We profiled genome-wide gene expression using a 3′-biased RNA sequencing method, 3SEQ (*Beck et al., 2010*), in order to maximize SNP coverage in the 3′ UTRs, where the average SNP density is ~1 SNP per 170 bp (compared to 1 SNP per 265 bp in the coding sequence). Mapping of reads separately against the Cast (*Keane et al., 2011*) and B6 genomes allowed us to eliminate reference mapping bias (*Figure 2—figure supplement 1*) as the average fraction of reads supporting the B6 allele (B6 allele fraction, 'BAF') was 0.502. Overall, out of a possible 6917 genes with sufficient reads covering SNPs, 1594 (23%) showed significant ASE ($p < 0.01$, paired t-test, Bonferroni corrected; *Figure 2A,B*). We next backcrossed F1 mice (derived from female B6 by male Cast crosses) with B6 parentals to obtain N2 embryos. We harvested 154 N2s at E11.5, performed 3SEQ, and quantified ASE in the approximately 50% of embryos that were heterozygous at each gene. There was excellent agreement between average BAF estimates from F1s and N2s (*Figure 2C*; rho = 0.77).

To quantify developmental progression for each embryo at the time of harvest, we counted the number of somites, which form every 90–120 min (*Saga and Takeda, 2001*) and ranged from 42 to 57 in our embryos. Of the 7712 genes with moderate expression of at least 5 reads per million (RPM), 1796 genes showed significant correlation between total expression and somite number ($q < 0.05$, Spearman correlation). Genes changing over this developmental time course were enriched for

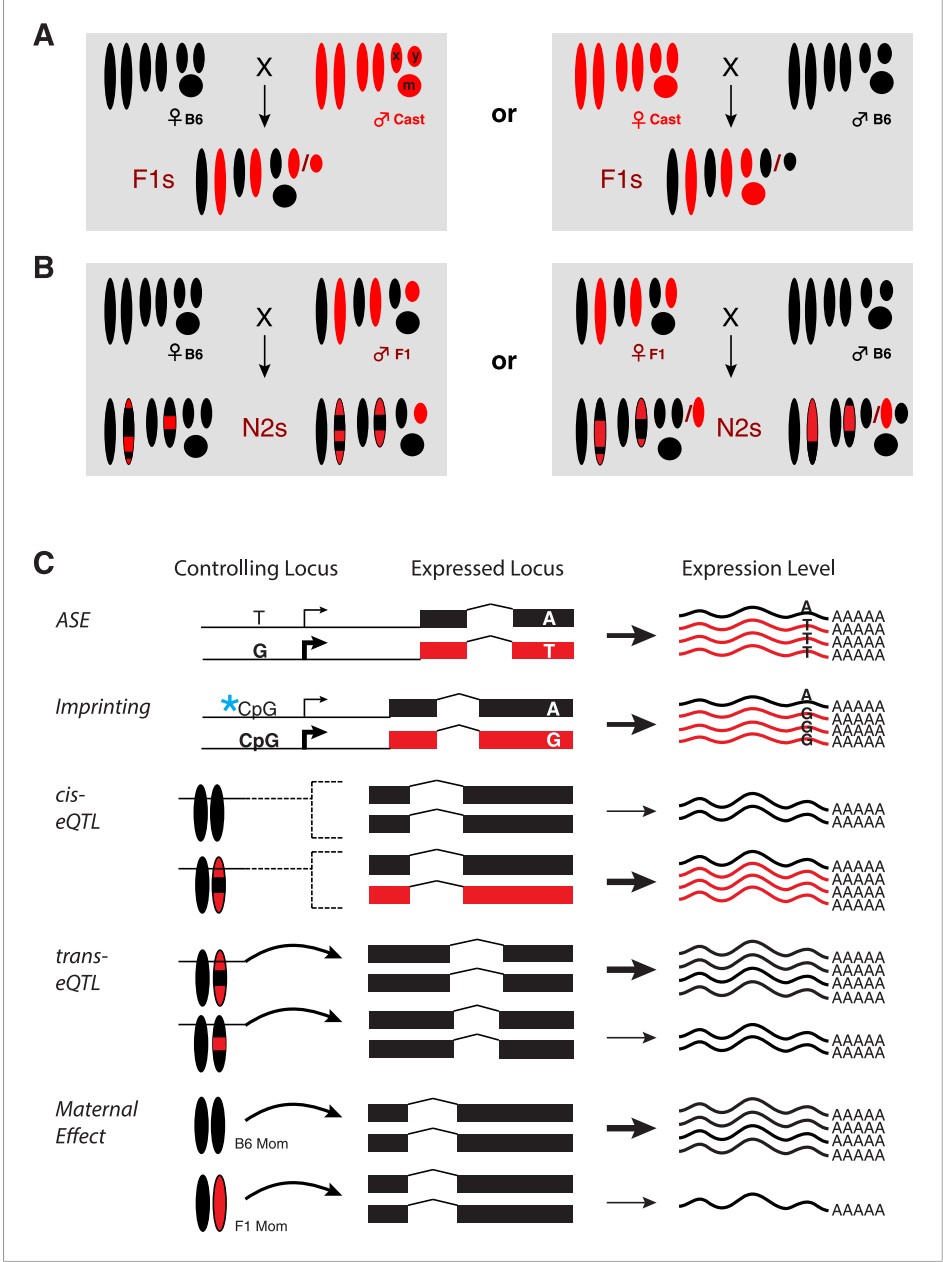

**Figure 1**. Experimental design. (**A**) Reciprocal crosses of B6 and Cast parents were used to generate F1 hybrid embryos. (**B**) F1 hybrid adults were backcrossed to B6 to generate N2 embryos via reciprocal crosses of B6 females with F1 males and F1 females with B6 males. Chromosomal regions inherited from a B6 parent are black and Cast regions are red. (**C**) Example genotypes at a controlling genetic locus, its corresponding target gene, and resulting gene expression levels for different types of gene regulatory variation assayed. Allele-specific expression (ASE) and imprinting are measured by allele-specific read counts, whereas other types measure total gene expression levels. X: X chromosome; Y: Y chromosome; m: mitochondrial DNA.

a number of gene ontology (GO) categories indicative of important developmental processes occurring at E11.5, including synaptogenesis (q = 2.2 × 10$^{-8}$, GOrilla) and eye lens formation (q = 5.8 × 10$^{-7}$). A third significant GO category, blood microparticles (q = 9.6 × 10$^{-7}$), includes the globins, which are beginning to undergo the transition from fetal to adult form over the developmental time period we assayed (*Noordermeer and de Laat, 2008*) (*Figure 2D*).

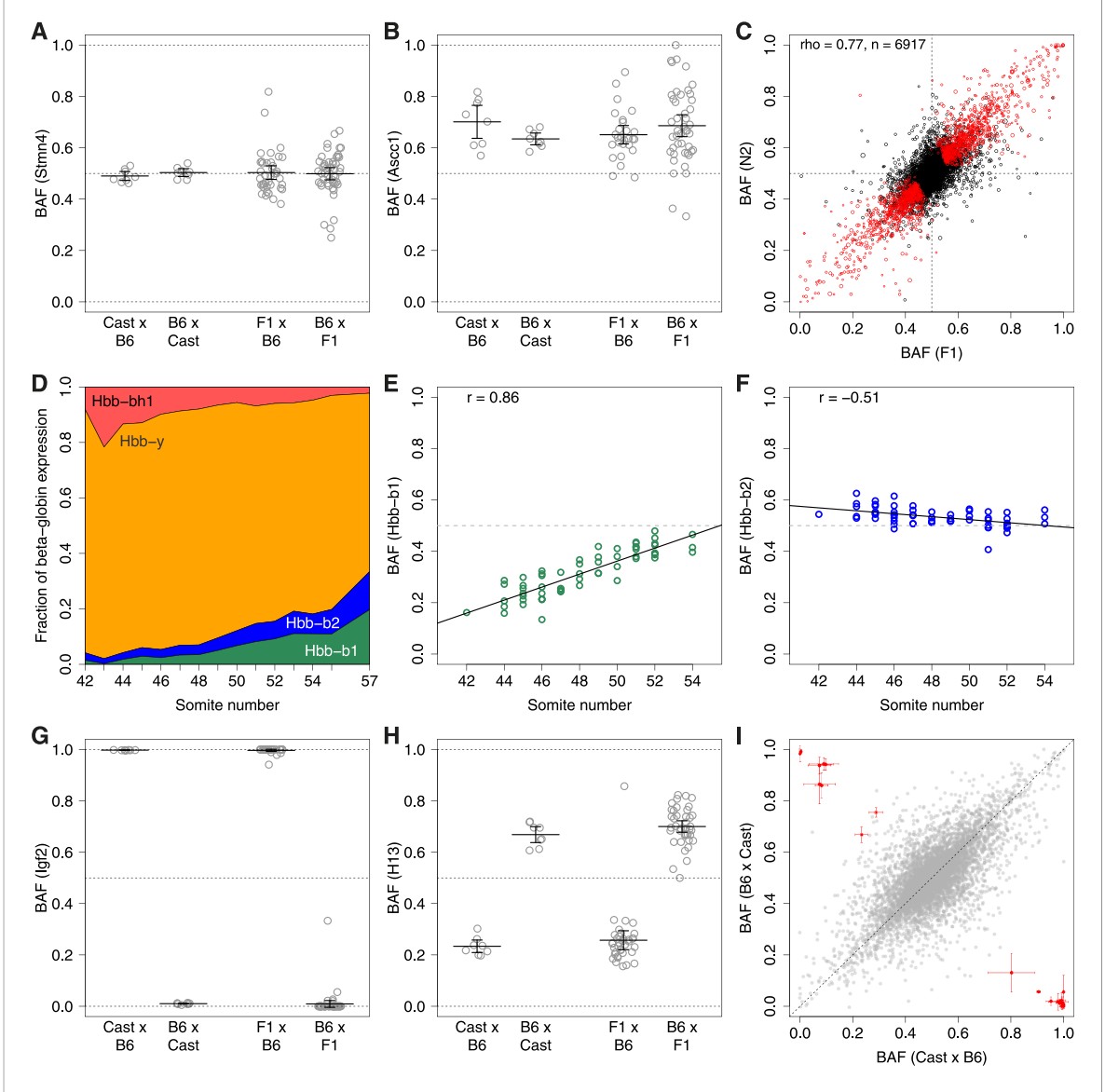

Figure 2. ASE. B6 allele fraction (BAF) for (A) non-ASE gene *Stmn4*, (B) ASE gene *Ascc1*. Each point represents a single embryo, grouped by cross (F1 embryos on left, N2 embryos on right). Cross mother is listed first. Bars are mean BAF and 95% confidence intervals. (C) Correlation of average BAF between F1 and N2 embryos. Significant ASE genes (called using union of F1 and N2 data) are red. Point size corresponds to relative expression level. (D) Transition from fetal (*Hbb-bh1*, *Hbb-y*) to adult (*Hbb-b2*, *Hbb-b1*) beta-globins, shown as fraction of total expression of all beta-globins, averaged across individuals by time point (somite number). (E and F) Divergent temporal regulation for B6 and Cast alleles of *Hbb-b1* (E) and *Hbb-b2* (F) in heterozygous N2 mice. (G and H) BAF for imprinted genes *Igf2* (G) and *H13* (H). (I) Significantly imprinted genes (red, with 95% confidence intervals; n = 31) clearly separate from non-imprinted genes (gray, n = 8261; *Figure 2—figure supplement 2*).

The following figure supplements are available for figure 2:

**Figure supplement 1**. B6 allele fractions.

**Figure supplement 2**. Age-dependent ASE.

**Figure supplement 3**. Significantly imprinted genes.

By correlating ASE with somite number, we identified twelve genes that show significant differential regulation between B6 and Cast alleles as a function of developmental timing (q < 0.10, Spearman rho). Globins and functionally related genes were well represented in this set, which included both major (*Hbb-b1*) and minor (*Hbb-b2*) adult beta globins. Both genes increase in absolute level and relative expression compared to fetal beta globins (*Figure 2D*; *Figure 2—figure supplement 2*), but the B6 allele of the major form is delayed compared to the Cast allele (*Figure 2E*; rho = 0.86, q < $10^{-14}$), whereas the opposite occurs for the minor form (*Figure 2F*; rho = −0.51, q = 0.02). By the end of our time course, the alleles are equally expressed (*Figure 2E,F*). Transferrin also increases in absolute expression but the Cast allele is delayed compared to the B6 allele (rho = −0.46, q = 0.02; *Figure 2—figure supplement 2*). These results suggest that compensatory mechanisms have evolved in one or both strains to keep the overall regulatory dynamic of oxygen physiology stable during this important switching period.

Comparison of embryos from either side of the reciprocal cross allowed us to identify parent-of-origin dependent ASE patterns indicative of imprinting. Consistent with previous studies in mouse embryos (*Babak et al., 2008*), we found a total of 31 genes with significant imprinting (*Figure 2G–I*; p < 0.01, binomial test, Bonferroni corrected), all of which had been previously identified. We note that the clear separation of most of the imprinted loci from the noise of the bulk of all expressed genes, with few exceptions (*Figure 2I*; *Figure 2—figure supplement 3*), suggests that most imprinting generally acts uniformly across the embryo, as expected from the methylation-based mechanism that sets the expression state of the regulated allele pre-zygotically (*Reik et al., 1987*).

Imprinting is not the only molecular mechanism that results from the mother–offspring resource allocation conflict in mammals (*Moore and Haig, 1991*). To explore other genetically-encoded parental effects, we took advantage of the backcross design, in which N2 offspring have genotypically distinct parents. We identified a large number (n = 331) of embryonically expressed genes whose total expression levels differed significantly depending on the genetic background of the mother (Mann Whitney U test, p < 0.01, Bonferroni corrected), irrespective of the genotype of the embryo (*Figure 3A,B*; *Supplementary file 1*). For example, *Angptl4* expression increases 52% from an average of 107 RPM in embryos with an F1 mother to an average of 163 RPM in embryos with a B6 mother (*Figure 3C*). As for the magnitude of the effect, these maternal effect target genes exhibited up to a 3.5-fold (e.g., *Polm*: 2.5–11-fold at a 95% confidence interval) difference in total expression level depending on genotype of the mother.

We were able to exclude a number of alternative mechanisms to explain these maternal effects. All F1s used for the backcross were derived from B6 female by Cast male matings, and therefore all N2 embryos carry the B6 mitochondrial genome, ruling out a confounding mitochondrial effect. The maternal effects were unchanged when considering sex, X chromosome genotype, litter, and developmental stage. Furthermore, for the vast majority of genes with SNP coverage, we were able to verify that maternal effect target genes were not imprinted or the result of contamination by maternally expressed transcripts. We found that maternal genotype by itself was a better predictor of the maternal effects than maternal genotype in conjunction with embryonic genotype at any of the imprinted loci, indicating that the maternal effects were not the result of trans effects downstream of imprinted genes. To exclude the possibility of a batch effect in our 3SEQ data, we used quantitative RT-PCR (qPCR) to confirm 4 genes as maternal effect target genes.

The most significant of the maternal effect target genes, *Angptl4* (*Figure 3C*; p = 8 × $10^{-13}$ by 3SEQ, p = 1 × $10^{-11}$ by qPCR), functions in lipid metabolism and energy homeostasis, suggestive of a role in responding to the placental nutrient supply (*Yoshida et al., 2002*). Similarly, *Gpx3* (*Figure 3B*; p = 5 × $10^{-12}$ by 3SEQ, p = 6.6 × $10^{-4}$ by qPCR), a glutathione peroxidase, is an antioxidant enzyme that acts in fat and glucose metabolism and has been associated with obesity and type 2 diabetes in mouse and human (*Lee et al., 2008*; *Chung et al., 2009*). Both of these genes are transcriptionally regulated by the peroxisome proliferator-activated receptor PPARγ.

To examine how genetic variation carried by the embryo itself affects gene expression phenotype, we performed eQTL analyses by correlating expression levels of each measurably expressed gene ('target') with N2 embryo genotypes (either B6/Cast or B6/B6) at each recombination-defined locus ('controlling locus'). Genotyping and recombination analysis based on the expressed SNPs (*Figure 4A*) divided the autosomes into 986 marker region blocks, each separated by one or more recombination events that define candidate controlling loci. We tested every controlling locus for

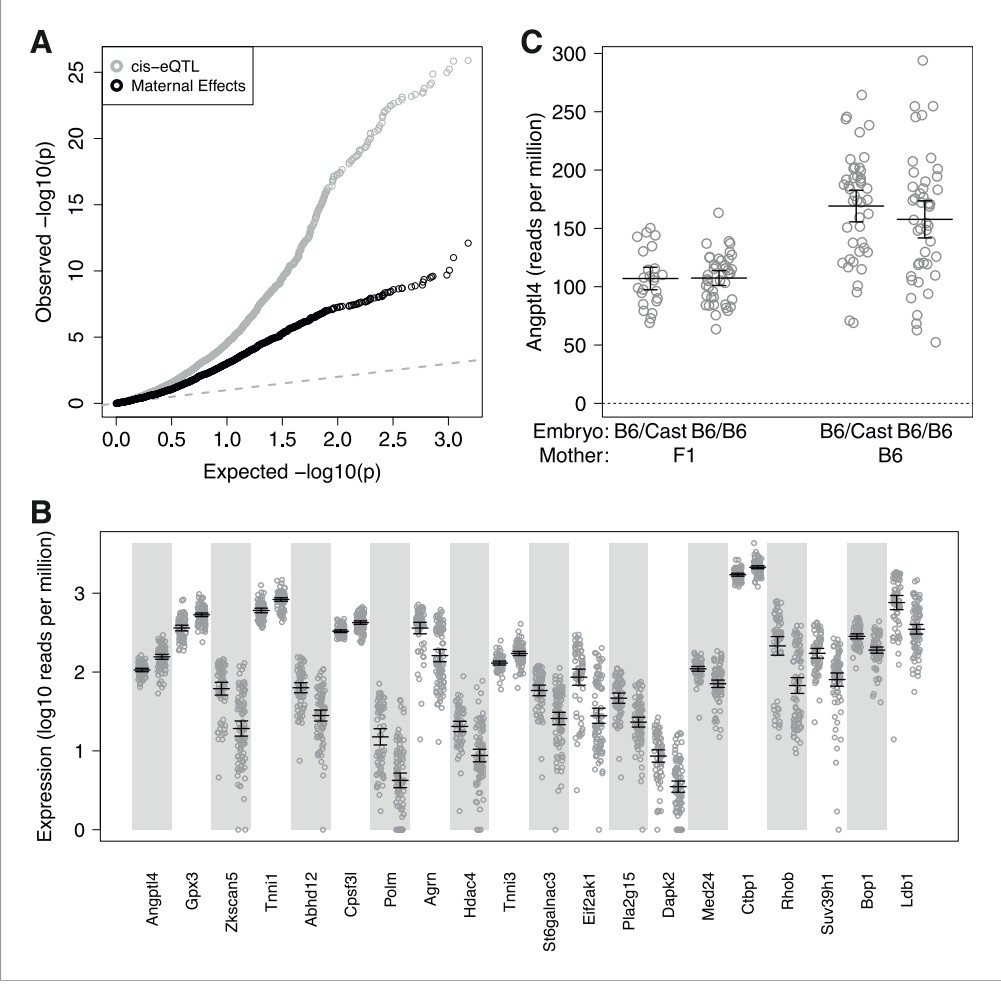

**Figure 3**. Maternal effect target genes. (**A**) Quantile–quantile (qq) plot for maternal effect target genes (black circles), compared to cis-expression quantitative trait loci (eQTL) (gray circles). (**B**) Expression by embryo for the top 20 maternal effect target genes, separated by maternal genotype. For each gene, samples are grouped by F1 mother (left) and B6 mother (right). (**C**) Total expression for *Angptl4* for embryos grouped by embryonic genotype and maternal genotype. Error bars indicate mean and 95% confidence intervals.

correlation with gene expression of every gene. At a genome-wide significance level of q < 0.01, we identified a total of 1034 eQTL.

1017 of these eQTL (98%) mapped close to the target gene (*Figure 4B*), suggesting cis-regulation as the underlying mechanism. A gene's difference in expression between heterozygotes and homozygotes could accurately predict its allelic ratio as assayed by ASE (*Figure 4C*). This high correlation (rho = 0.58, n = 4821; rho = 0.70 when considering only the n = 1000 genes with the highest allele-specific read coverage) provides a systematic validation of the large effect of embryonic cis-regulatory polymorphisms on expression phenotype. Note that our study has sufficient power to identify substantial numbers of eQTL when performing this all–loci–by–all–genes analysis. If, instead, we were focused specifically on only studying cis-regulation, we could have increased our power to identify cis-eQTL by only comparing genes to their nearby loci, thereby reducing the number of statistical tests being performed.

In contrast to the large number of cis-eQTL, we found few trans-eQTL regardless of FDR cutoff (*Figure 5A*). After the top 3 (each comprising a controlling locus–target gene pair, e.g., *Figure 5B*), significance drops off rapidly compared to background, unlike for the cis-eQTL (*Figure 5C*). To explore this lack of trans effects further, we reasoned that a set of genes with a high prior probability of having downstream effects might be enriched for trans-controlling loci. Our cis-eQTL/ASE loci

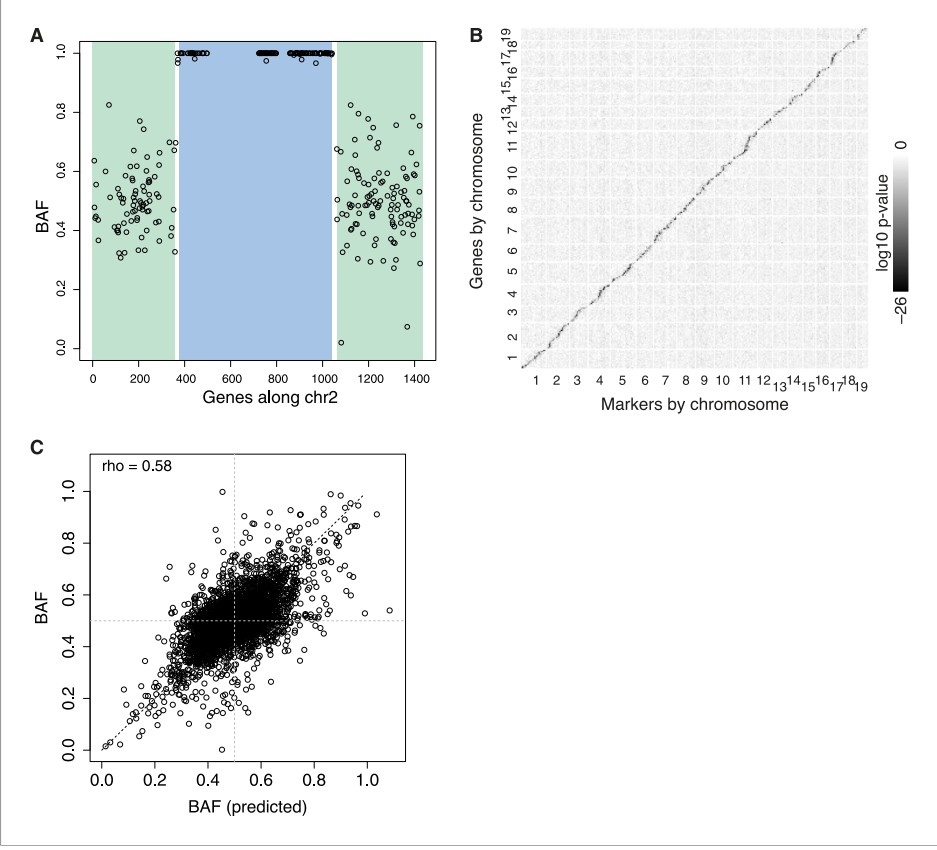

**Figure 4**. eQTL. (**A**) Example N2 embryo data used to call recombination events for chromosome 2. Each point represents BAF for a single gene. The HMM calls green regions as heterozygous, blue regions as homozygous, with two recombination events occurring in the short intervening white regions. Only genes with sufficient read coverage of single nucleotide polymorphisms (SNPs) are shown as points; genes with under 20 SNP reads are still genotyped, based on neighboring genes. (**B**) Overview of all eQTL, arranged by genomic position, shaded by log10 p-value. cis-eQTL, along the diagonal, show the only appreciable signal. (**C**) Predicted BAF from transformed expression levels in heterozygotes and homozygotes (x-axis) correlate with BAF measured by ASE (y-axis).

The following figure supplement is available for figure 4:

**Figure supplement 1**. Random forest analysis.

comprise a large set of such candidates because differences in expression levels of a gene product may propagate to downstream targets (*Yvert et al., 2003*). However, we found that neither strong cis-eQTL nor ASE loci are enriched for trans-eQTL. Finally, application of a random forest model (*Michaelson et al., 2010*; *Francesconi and Lehner, 2014*) that simultaneously considers developmental progression (number of somites) and genotype also did not enrich for trans-eQTL (*Figure 4—figure supplement 1*). We estimate that the number of comparable trans-eQTL is about 1% that of cis-eQTL.

Because a trans-regulatory effect is potentially distributed over several target loci, we asked whether there was evidence for 'hotspots', or controlling loci whose genotypes correlate with small expression differences at numerous target genes (*Brem et al., 2002*; *Schadt et al., 2003*). Permutation tests revealed a significant signal (p < $10^{-24}$ KS test, *Figure 5D*) in support of hotspots, but no specific candidates. Reinforcing the above result that ASE genes do not act as trans-controlling loci, strong ASE genes were not substantially more likely to act as hotspots than weaker ASE genes (*Figure 5D*; p = 0.20, KS test). Therefore, at this stage of embryogenesis, we found no evidence for downstream propagation of genetically controlled expression level differences.

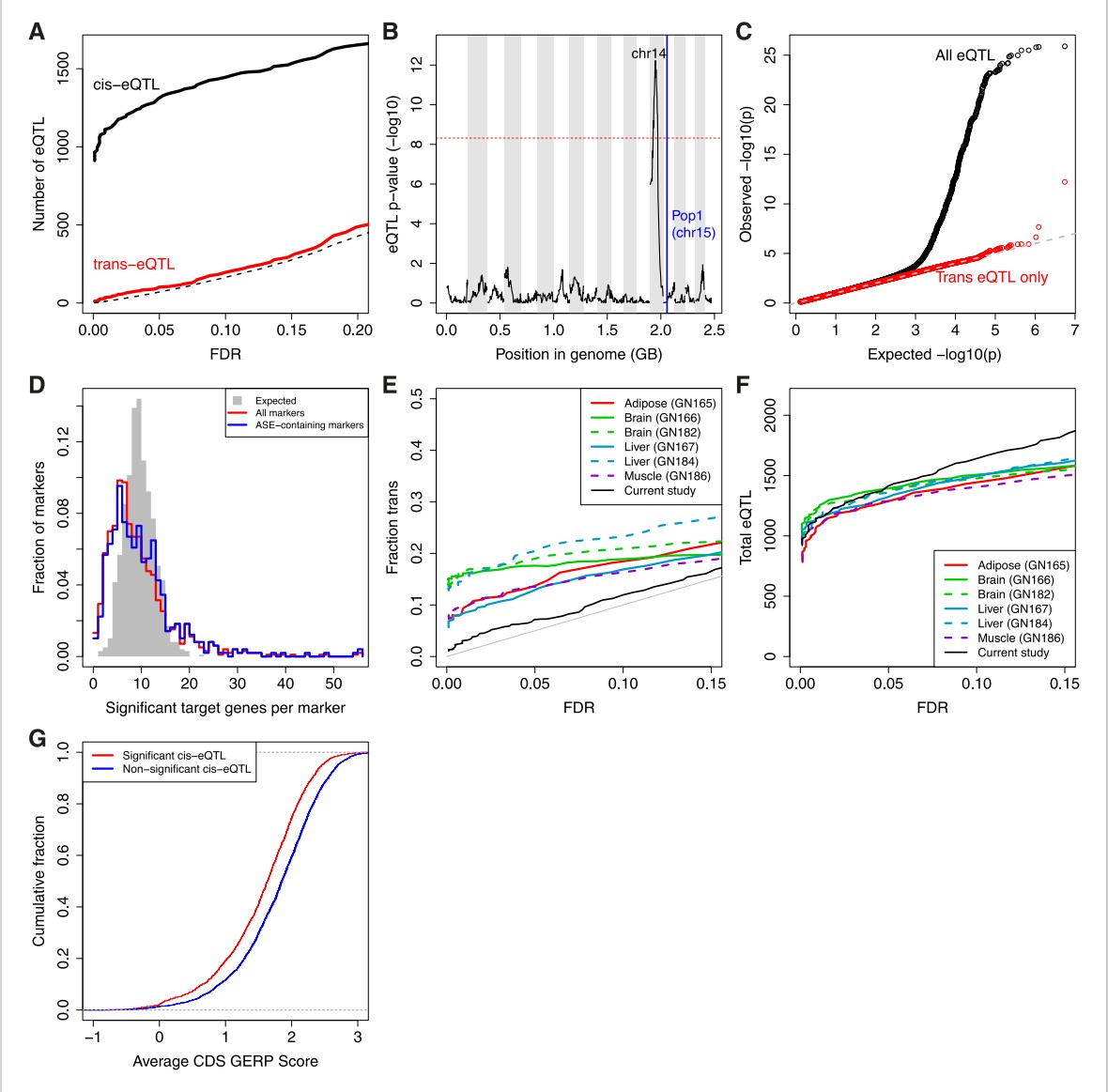

**Figure 5**. Trans-eQTL. (**A**) Number of cis-eQTL (black) and trans-eQTL (red) as a function of FDR. Dashed line indicates number of false positive eQTL expected. (**B**) eQTL for *Pop1*, the most significant trans-eQTL target (vertical blue line). Candidate controlling loci are shown across the x-axis in genome order; only the peak on Chr14 is significant. Chromosomes are shaded in alternating background colors. (**C**) qq plot for all gene–marker pairs (black) and gene–marker pairs only in trans (red). Gray straight lines are y = x, x axis is expected p-values based on permutations. (**D**) Trans-eQTL hotspots, shown as the distribution of significant target genes per marker region (p < 0.005) across all marker regions (red) or only marker regions containing the top 50% of ASE genes (blue). Gray shows the expected distribution if there were no hotspots. (**E** and **F**) Comparison of our study with that of eQTL in adult tissues as a function of FDR. (**E**), Trans-eQTL as a proportion of total eQTL. (**F**), total number of eQTL. GeneNetwork sample ID indicated in parentheses. (**G**) Evolutionary constraint (GERP score) in coding regions (CDS) for the most significant (red; n = 1347) and non-significant (blue; n = 3726) cis-eQTL genes.

We hypothesized that the paucity of trans-eQTL was due to increased evolutionary constraint of regulatory networks during embryonic development. Since trans-eQTL have previously been found to have weaker effects than cis-eQTL (*Montgomery and Dermitzakis, 2011*), an alternative hypothesis is that we were simply underpowered to find trans-eQTL. To distinguish these possibilities, we obtained genotype and expression data from previously published eQTL studies performed in adult mouse tissues (*Wang et al., 2006*; *Yang et al., 2006*; *Chen et al., 2008*). After sampling individuals to match the same number of total eQTL found at an FDR of 0.05, we found that all previously published studies conducted in adult tissues showed a substantially higher proportion of trans-eQTL than did our

embryos (*Figure 5E,F*). This result was robust to varying the sample size from the previously published data, to whether p-values or FDR was used to assess significance, and to varying the thresholds for comparing proportions of trans-eQTL across the ranked eQTL. Based on this result and our high power to detect cis-eQTL even at low FDR, we conclude that embryonic trans-eQTL are subject to elevated selective constraint compared to adult tissues.

If cis-regulatory divergence in embryos is strongly buffered and does not lead to downstream trans-regulatory effects, why does embryogenesis tolerate such a large number of cis-effects in the first place? We hypothesized that divergent cis-regulation between strains occurs preferentially in genes with lower evolutionary constraint. In support of this hypothesis, ASE genes were significantly depleted of GO categories related to organ development ($p = 6 \times 10^{-5}$, $q = 0.05$), as well as gene regulation ($p = 1 \times 10^{-4}$, $q = 0.07$), and transcription factors ($p = 3 \times 10^{-5}$, $q = 0.03$), demonstrating constraints on variation in core regulatory genes. Furthermore, we found that evolutionary constraint at the DNA sequence level (*Cooper et al., 2005*; *Davydov et al., 2010*) was lowest in the most significant cis-eQTL (*Figure 5G*; rho = −0.14, $p = 6 \times 10^{-13}$; n = 2788). Similarly, ASE magnitude of effect correlated negatively with evolutionary constraint (rho = −0.10, $p = 9 \times 10^{-8}$, n = 2612), whereas cis-eQTL in adult mouse tissues showed no correlation (|rho| < 0.05). Thus, the strong enrichment for cis-eQTL/ASE in less conserved genes is unique to embryos.

## Discussion

We have performed the first, to our knowledge, comprehensive global analysis of the effects of genetic variation on mammalian gene expression during embryogenesis. Our experimental design allowed us to simultaneously assay the effects of genetic variation on imprinting, maternal effects, cis-regulation, and trans-regulation. Supporting the high quality of our data and analytical methods, we found good agreement between related measurements of cis-regulatory variation when comparing ASE in F1 embryos (sequenced at high coverage) with ASE in N2 embryos (sequencing many individuals), and when comparing ASE with cis-eQTL.

Our two mouse strains exhibit SNPs at 17.7 million sites, about sixfold more than the heterozygosity in African humans (*1000 Genomes Project Consortium, 2012*). The vast majority of these polymorphisms have accumulated as a result of genetic drift (*Kimura, 1983*) over the time since the present-day European and Southeast Asian *M. musculus* populations were separated from their common ancestral population. Of the small subset of polymorphisms that affect molecular function, some may produce an advantageous phenotype subject to positive selection, some are deleterious to organismal fitness and are therefore subject to purifying selection, and yet others are selectively neutral (subject to drift). Missense polymorphisms are severely underrepresented genome-wide and exhibit lower derived allele frequencies relative to other genomic sites (*1000 Genomes Project Consortium, 2012*), reflecting the global predominance of purifying selection. This signal of purifying selection is weaker but still substantial for noncoding gene regulatory sites (*ENCODE Project Consortium, 2007*).

Given these known dynamics, our study sheds light on the microevolution of gene regulation. Gene expression divergence of the type we quantified in this work occurs as a result of polymorphisms in regulatory regions or in expressed regulatory gene products. The former manifests as cis-effects, the latter as trans-effects. Because we focused on embryonic gene expression, two competing hypotheses as to what patterns we might find were plausible. Under the first hypothesis, the substantial DNA sequence divergence between mouse strains would be reflected in many expression differences. The alternative hypothesis is that few expression changes would be detected between strains. This hypothesis is suggested by the developmental hourglass model (*Domazet-Lošo and Tautz, 2010*; *Kalinka et al., 2010*), which posits that gene expression patterns are most highly conserved in the middle of embryonic development (the so-called 'phylotypic stage').

In support of the first hypothesis, we found substantial numbers of significant eQTL, demonstrating that a sizeable subset of all segregating polymorphisms causes expression changes. However, we found that nearly all acted in cis, rather than trans. This depletion of trans-regulatory variation is substantially stronger than that observed across several adult mouse tissues. In contrast to adult tissues, eQTL acting in our embryos were correlated with lower DNA sequence conservation, supporting the conclusion that evolutionary constraint on gene expression regulation is markedly stronger during mammalian embryogenesis than in the adult. In addition, the identified cis-expression changes do not themselves cause detectable trans-effects. These findings support the idea of the

developmental hourglass model (*Domazet-Lošo and Tautz, 2010*; *Kalinka et al., 2010*) while suggesting that substantial neutral divergence of molecular phenotype (gene expression) can accumulate in the form of cis-regulatory variation even under strong evolutionary constraint at the organismal (embryonic) level.

Previous studies have shown enrichment for trans-regulation when using an eQTL model that considers both genotype and stage during *C. elegans* larval development. Perhaps because of the nearness of our embryos to the mouse phylotypic stage, or because of fundamental differences between mice and worms, we found that incorporating developmental timing (i.e., somite number) into our eQTL model did not alter the balance between cis- and trans-regulation. Somite number was, however, correlated with significant changes in ASE, indicating differential developmental timing of gene expression between the Cast and B6 strains. In particular, we found that alleles of several hemoglobin-related genes, including the adult beta globins and transferrin, were expressed at differing levels between the strains at early stages of the fetal-to-adult hemoglobin transition, but then equalize by the end of our developmental time course. These results demonstrate our ability to identify subtle tissue-specific expression differences between strains, despite sequencing from whole embryos containing a mix of tissues.

It is formally possible that some of the significant eQTL identified here act in trans, even though their target genes are located in genomic proximity. However, the good agreement between eQTL and ASE suggests that the majority of our local eQTL are indeed cis-eQTL. Furthermore, because there is no reason to believe that the proportion of locally-situated but trans-acting eQTL is higher in embryos than in adults, it does not materially alter our conclusions if a small number of our local eQTL do act in trans.

Because of the large number of embryos collected and individually sequenced for our study, we were able to accurately quantify ASE mean and standard deviation for thousands of genes. When stratifying embryos by side of cross, we found that nearly all imprinted genes showed very strong allelic bias, meaning that the epigenetically silenced allele is expressed at only low levels across the embryo. Consistent with previous studies that found few genes exhibiting tissue-specific imprinting [apart from those found in the extra-embryonic tissues; reviewed in *Prickett and Oakey (2012)*], we observed no weak allelic bias that might result from averaging effects across tissues.

In addition to the clear signature of parent-of-origin-dependent allelic expression, we also found strong maternal effects on total gene expression levels for hundreds of genes. Whereas most previous studies have employed F1 intercrosses in which parental genotypes are indistinguishable, we used a backcross design, enabling us to track the potentially important sources of genetic variance contributed by the parents. The observed maternal effects are restricted to pre-natal influences and separate from imprinting and mitochondrial effects.

Our results demonstrate that the genotype of the mother plays an important role in regulating gene expression, not just after birth, as has been previously suggested, but also during embryogenesis, affecting the expression of hundreds of genes (*Bult and Lynch, 1997*; *Weaver et al., 2004*; *Wolf et al., 2011*). The implications of this finding are significant for human genetic twin, familial, and genome-wide association studies. What is usually thought of as familial environment [e.g., *Grundberg et al. (2012)*] may have a substantial heritable genetic component exerting an effect not on the generation carrying it but on the offspring.

Parental genetic influences on offspring phenotype have been implicated in growth regulation and adult-onset obesity- and diabetes-related diseases but the gene targets of such effects have not been identified (*Jarvis et al., 2005*). While we cannot rule out an epigenetic or RNA-mediated mechanism underlying this regulation, it is more likely acting through the placenta's role of transmitting nutrients, hormones or other factors to the embryo. This hypothesis is supported by the known functions of the most significant target gene, *Angptl4*, which responds to lipid metabolism and energy homeostasis, and participates in angiogenesis (*Koster et al., 2005*; *Scott et al., 2012*). We hypothesize that these maternal effect target genes continue to be expressed differently depending on maternal genotype, and may be at least partly responsible for the previously observed effect of maternal genotype on pre-natal growth (*Cowley et al., 1989*; *Wolf et al., 2011*).

In summary, using our hybrid ASE/eQTL approach we were able to assay all four types of genetic variation in gene expression regulation. We found that cis-regulatory variation caused the largest divergence in overall gene expression in mouse embryos, affecting over 1000 genes with up to sevenfold differences in expression levels between the strains. These cis-regulatory variants did not translate into trans-regulatory divergence, and we identified only ~10 trans-eQTL genomewide.

Imprinting affected the allelic balance of dozens of genes, generally resulting in complete or nearly complete silencing of the inactive allele. Finally, we identified over 300 maternal effect target genes. The number of maternal effect targets, and their overall significance, are only slightly more modest than cis-regulatory effects, but nonetheless substantial, with approximately 1/3 as many genes involved, and maximal effect size of up to 3.5-fold difference or approximately 1/2 as large. Given this context, it is clear that maternal effects can play a substantial genetic role in determining embryonic gene expression patterns. We speculate that maternal effects contribute to evolutionary divergence, allowing expression heterogeneity in the embryo to build as a result of differences in the adult mother, where gene regulatory networks show less evolutionary constraint than in the developing embryo.

## Materials and methods

### Mouse strains and crosses

6-week old mice were purchased from The Jackson Laboratory. All 'Materials and methods' were carried out in accordance with the Administrative Panel on Laboratory Animal Care protocol and the institutional guidelines set by the Veterinary Service Center at Stanford University.

The inbred strain C57BL/6J and wild-derived inbred strain CAST/EiJ were used in reciprocal crosses (*Figure 1*). C57BL/6J females were crossed with CAST/EiJ males to generate (B6 × Cast)F1 hybrid embryos (n = 8), and CAST/EiJ females were crossed with C57BL/6J males to generate (Cast × B6)F1 hybrid embryos (n = 8). (B6 × Cast)F1 progeny were backcrossed to C57BL/6J in reciprocal crosses to generate 154 N2 embryos (F1 × B6 = 65 and B6 × F1 = 89). Individual embryos from 2 litters from each side of the F1 cross (2 B6 × Cast and 2 Cast × B6), and 12 F1 × B6 litters and 18 B6 × F1 litters from the N2 cross were analyzed.

For all crosses, following identification of a vaginal plug on the morning after mating (E0.5), males were removed; pregnant females were housed separately.

Whole embryos were harvested at E11.5 and dissected to remove all embryonic membrane and placental material. Somite numbers were recorded for each embryo at the time of dissection.

### Library preparation and sequencing

To isolate total RNA, individual embryos were homogenized in Trizol reagent (GibcoBRL/Invitrogen, Grand Island, NY) and RNA was extracted per manufacturer's directions. Poly(A)$^+$ RNA was isolated using the Qiagen Oligotex Mini kit and 3SEQ libraries were constructed as described (*Beck et al., 2010*). For the N2 embryos, oligo(dT) and PCR primers were barcoded with Illumina multiplexing sequences. F1 data were produced in several sequencing batches, first on an Illumina GAIIx sequencer producing 76 bp or 80 bp reads, then on an Illumina HiSeq 2000 sequencer producing 101 bp reads. N2 data were produced only on the HiSeq 2000. Libraries from 12 individually barcoded N2 samples were pooled in equimolar amounts and sequenced in a single flow-cell lane to produce single-end 101 bp reads.

### Read preprocessing

Prior to read mapping, 3′ poly(A) sequences likely to be derived from the mRNA poly(A) tail were trimmed using a procedure that tolerated some amount of sequencing error: we computed a score $s_i$ for each position $i$ in the read which was equal to the sum of the quality scores of each A after position $i$ minus three times the sum of the quality scores of each non-A after position $i$. All bases after the highest-scoring position were trimmed if this would result in trimming at least 3 bases.

The first 18 bases and the last 7 bases of the HiSeq F1 reads were trimmed to make them the same length as the GAIIx reads. This was done to avoid read length being a confounding factor when analyzing ASE in F1. Only poly(A) stretches were trimmed from the N2 data.

### Read alignment and filtering

A *M. m. castaneus* genome was created in silico by applying the high-quality Cast SNPs (*Keane et al., 2011*) to the reference C57BL/6 genome mm9. For B6, we used the mm9 genome. For each of these two reference genomes, a transcriptome was computed using RefSeq annotations, including spliced transcripts of all genes separated by 200 nucleotides, and each spliced transcriptome was added to its reference genome sequence. Bowtie2 (*Langmead and Salzberg, 2012*) was used to map all reads to the composite genomes (Cast or B6 including spliced transcripts) with settings '-k 200 –score-min

L,0,-0.10 –rdg 2,2 –rfg 2,2 –mp 1,1'. These settings recovered up to 200 sub-optimal genomic hits, allowing us to stringently filter reads derived from repetitive regions of the genome as follows: If the difference in edit distance (1 for each mismatch and 2 for each indel) between a read's best alignment and its second-best alignment was within 3, the read was discarded. Reads were also filtered out unless all best alignments in the composite genome corresponded to the same genomic position (e.g., if a read aligned equally well to an intergenic region and to a splice junction in the transcriptome). Each aligned read was then consolidated into a single genomic alignment, including intronic splice gaps.

After filtering, each F1 dataset had an average of 15.8 million mapped reads, and each N2 dataset had an average of 1.2 million reads.

## Expression quantification

Gene expression was quantified using only reads mapping to exons. Read counts were typically normalized by the total number of sequenced reads per library, multiplied by $1 \times 10^6$, resulting in RPM. For the eQTL analyses, genes were removed if over 5% of the mapped reads were filtered based on the second-best hit criteria above. Genes were also discarded if they matched the retrotransposed genes list compiled by UCSC (*Kent et al., 2002*). In the F1s, there were 13,574 genes with detectable expression (13,469 across N2s), 7554 of which passed the analysis threshold of 5 RPM (7712 in N2s).

We found *Xist* expression followed a bimodal distribution, and used this to identify females (robust *Xist* expression) and males (little or no *Xist* expression). All 5 genes significantly different between males and females, including *Xist* and its antisense transcript *Tsix*, were located on the X or Y chromosomes. At E11.5, embryos are just beginning sexual differentiation, so the small number of sex-specifically expressed genes is unsurprising. Because of the small number of affected genes, as well as the repetitiveness of the sex chromosomes, subsequent analyses were restricted to genes and marker loci on autosomes.

## Genotyping

Genes were assigned a preliminary genotype based on the proportion of B6-supporting reads, either 'unknown' if <20 SNP-covering reads were found, or homozygous or heterozygous otherwise. For each chromosome, preliminary genotypes were then fed through a Hidden Markov Model (HMM) that penalized transitions between the heterozygous and homozygous states. The HMM in effect smoothed the genotypes and allowed us to infer genotypes for unknown genes falling between confidently genotyped genes. A post-processing step assigned the 'unknown' genotype to genes with low read coverage falling near putative recombination sites, typically affecting only a small number of genes for each chromosome.

We combined adjacent genes with identical genotypes across all embryos into a total of 986 marker regions across the 19 autosomes. A more aggressive combining, allowing up to 5 recombination events across all N2 embryos, resulted in 286 marker regions. Downstream analyses used the consensus genotype in each embryo, assigning the unknown genotype where the number of homozygous and heterozygous embryos was close to equal.

## GO

GO enrichment analyses were performed with GOrilla (*Eden et al., 2009*), which enables easy visual inspection of results, and depletion analyses were performed using a hypergeometric test. Only expressed genes (or those with allele-specific read coverage) were considered.

## ASE

To avoid the possibility of small insertions and deletions causing misalignment around SNPs, ASE was quantified using only high quality annotated SNPs that were at least 50 bp from an annotated Cast indel of any quality, based on the published mouse polymorphism resource (*Keane et al., 2011*). For reads covering multiple SNPs, each SNP received 1/n votes for a given genotype (Cast, B6 or other), to account for the low percentage of SNP positions with sequencing errors. For example, a read covering 3 SNPs, 2 of which match B6 alleles and 1 that matches neither the B6 nor Cast allele, would contribute 0.67 read votes to the B6 expression total. A read covering only a single SNP matching the Cast allele, would contribute a full 1 read vote to the Cast expression.

Because F1 and N2 ASE estimates largely agreed, ASE significance was estimated from the 170 pooled samples. Significance was assessed by subtracting the B6 read votes from the Cast read votes,

and then normalizing the differences based on library-size. This was necessary because the F1 samples were sequenced to substantially more depth than the N2 samples.

A paired t-test was then applied to test for significance of these differences against 0. This simple approach takes full advantage of the magnitude of difference between the alleles as well as the variance in allelic expression across the large number of samples sequenced in this study. We found that p-values calculated from the paired t-test correlated well with those derived by resampling, but were somewhat conservative. Hence, our estimates of ASE are likely to be slight underestimates. Additionally, genes were only called significant if their average BAF was less than 0.45 or greater than 0.55.

## Maternal effect target genes

Maternal effect target genes were identified by comparing expression normalized by library size from embryos with B6/B6 mothers to expression values from embryos with B6/Cast F1 mothers. We used a Mann–Whitney U test to calculate the significance of this difference.

Because all F1 mothers used for the N2 backcross were derived from the B6 × Cast side of the cross, all N2 embryos had B6 mitochondria. Maternal effects are similar in significance and direction when restricting analyses to only female embryos with two B6 X chromosomes, or when averaging expression across individuals in each litter. Furthermore, the effects are consistent across somite numbers, even for genes that change expression during development.

## qPCR

To exclude batch effects as a potential cause of maternal effects, we used qPCR to validate candidate maternal effect target genes on an orthogonal platform. 0.5 µg total RNA per sample was reverse-transcribed into cDNA using Oligo(dT)$_{20}$ primer and Superscript III (Invitrogen) in 20 µl reactions, according to the manufacturer's instructions. Following reverse transcription, each sample was treated with RNase H. 48.48 microfluidic dynamic array IFC chips (Fluidigm) were used to analyze the expression of 26 candidate maternal effect target genes and 15 control reference genes in 93 N2 samples. 1 µl of cDNA was pre-amplified using 2× Taqman PreAmp Master Mix (Lifetech) and 50 nM of each primer pair in 5 µl reaction volume, according to the manufacturer's instructions. The cycling program was 10 min at 95°C followed by 10 cycles of 15 s at 95°C and 1 min at 60°C. Following pre-amplification, each reaction was diluted fivefold. RT-qPCR on the dynamic array chips was conducted on the BioMark system (Fluidigm). 5 µl sample pre-mix containing 2.5 µl of SsoFast EvaGreen Supermix with Low ROX (Bio-Rad), 0.25 µl of DNA Binding Dye Sample Loading Reagent (Fluidigm) and 2.25 µl of diluted pre-amplification samples, as well 5 µl assay mix containing 2.5 µl of Assay Loading Reagent (Fluidigm), 2.25 µl EB Buffer (Qiagen) and 0.25 µl of 100 µM primer pairs (500 nM in the final reaction) were mixed on the chip using the IFC controller MX (Fluidigm). The thermal cycle was 60 s at 95°C followed by 30 cycles of 5 s at 96°C and 20 s at 60°C. A dissociation curve was also drawn for each primer pair.

The following eight control genes were chosen based on high consistency between Fluidigm chips: *Smarcc2, Rnf216, Nxf1, Cnpy3, Pmvk, Trmt1, Fam149b, Farp1*. These genes were used to normalize expression of the candidate maternal effect target genes using the standard curve method. Significance of the maternal effect was calculated using the Mann–Whitney U test on the normalized expression values using a Bonferroni-corrected p = 0.05 cutoff. Because of low consistency between Fluidigm chips, we were only able to validate four maternal effect candidate genes.

## eQTL

Between-sample read count comparisons showed some evidence of batch effects across N2 embryos, primarily at the level of sequencing lane. To control for these batch effects, we used PEER (*Stegle et al., 2012*) to normalize for both unknown covariates as well as the following known batch effects: embryo sex, sequencing lane, and somite count. For each gene and for each marker region, we performed a Wilcoxon rank-sum test comparing PEER-normalized expression in heterozygous and homozygous embryos. We found that the total number of significant eQTL was substantially higher using PEER-normalized read counts compared to simple library-normalized read counts, although the fraction of trans-eQTL was not significantly different. Results were also consistent when replacing the non-parametric ranksum test with the parametric t-test.

To remove the effects of correlated markers, downstream analyses used the most significant marker region on each chromosome for each target gene. To estimate false discovery rates, marker–sample assignments were permuted and eQTL were called using an identical procedure to the unpermuted data. Cis-eQTL were defined as those eQTL for which the marker region resides on the same chromosome as the target gene, and trans-eQTL were thus defined as the eQTL for which the marker region is on a different chromosome as the target gene. While this is a conservative definition of trans-eQTL, we found no evidence of trans-eQTL residing on the same chromosome as the target gene.

## ASE vs eQTL comparison

We estimated expected ASE ($BAF_{pred}$) from total expression levels using the following transformation for each gene:

$$BAF_{pred} = \frac{BAF_{exp}}{2 \times (1 - BAF_{exp})}, \quad \text{where } BAF_{exp} = \frac{\mu_{B6/B6}}{\mu_{B6/B6} + \mu_{B6/BCast}},$$

and $\mu_g$ are the mean total expression levels across all mice with genotype $g$ at the gene of interest. The $BAF_{pred}$ transformation is required to estimate $\mu_{Cast/Cast}$ from $\mu_{B6/B6}$ and $\mu_{B6/Cast}$. Because of variance in the estimation of $\mu_g$, the predicted BAF can be larger than 1.0.

## eQTL comparison with previous studies

We reanalyzed six previously published mouse eQTL datasets available from GeneNetwork (*Wang et al., 2003*). Array-normalized data were batch-normalized using PEER, then compared to genotype data provided by GeneNetwork. Because these previous studies used an intercross model with three possible genotypes, eQTL were called using a linear regression, analogous to the t-test in our backcross experiment.

## Random forests

eQTL were also mapped using a random forest method (*Michaelson et al., 2010*). We used the normalized read counts as input to the random forest, and included marker genotype, sex, side of the cross, sequencing lane, and number of somites as predictors. For 5 N2 embryos, no somite count was recorded, and hence the total number of input samples is 149 instead of 154. Genotypes were coded as 0.5 for heterozygous B6/Cast, 1.0 for homozygous B6/B6, and 0.75 for missing genotypes. False discovery rates were calculated as above using permuted input data after identifying the best markers per chromosome. We found results were similar when using the variable selection frequency method (*Michaelson et al., 2010*) or the mean decrease in prediction accuracy method (*Francesconi and Lehner, 2014*).

## Conservation

We found GERP scores broadly correlated among upstream, UTR, coding and downstream gene regions, and therefore show only the average coding sequence GERP score. We performed a Spearman correlation test to examine the hypothesis that GERP score is associated with cis-eQTL p-value and ASE effect size, although for clarity *Figure 5G* focuses only on the most extreme genes. Because the set of very lowly expressed genes is enriched for low GERP scores, we filtered out the lowest expressed genes in each tissue. We used RNA-seq data previously published (*Merkin et al., 2012*) for muscle, liver and brain to filter out lowly expressed genes in those tissue comparisons. The strength of the associations with GERP scores for our ASE and cis-eQTL as well as previously published cis-eQTL is robust to all but the most extreme changes in filtering cutoff.

## Acknowledgements

NS was supported in part by the Stanford Genome Training Program (NIH/NHGRI), JMB by the Biomedical Informatics Training Program (NIH/NLM), and JCB by the Burroughs Wellcome Fund Preterm Birth Initiative. We thank Elizabeth Finn for help with embryo dissections, Ghia Euskirchen and staff of the Stanford Center for Genomics and Personalized Medicine and Ziming Weng for help with sequencing, Cyril Ramathal and the Reijo-Pera lab for qPCR advice, and Casey Brown, Stephen Montgomery, and Eric Stone for discussions and comments on the manuscript.

All raw sequencing data are available in the Gene Expression Omnibus (http://www.ncbi.nlm.nih.gov/geo/) under accession number GSE62967.

## Additional information

### Funding

| Funder | Grant reference | Author |
|---|---|---|
| National Institutes of Health (NIH) | R01GM103787 | Julie C Baker |
| Burroughs Wellcome Fund (BWF) | | Julie C Baker |
| National Institutes of Health (NIH) | T32 HG000044 | Noah Spies |

The funders had no role in study design, data collection and interpretation, or the decision to submit the work for publication.

### Author contributions

NS, CLS, Conception and design, Acquisition of data, Analysis and interpretation of data, Drafting or revising the article; JMR, AS, Conception and design, Analysis and interpretation of data, Drafting or revising the article; JCB, Conception and design, Drafting or revising the article; SB, Analysis and interpretation of data, Drafting or revising the article

### Ethics

Animal experimentation: All experimental procedures were carried out in accordance with the Administrative Panel on Laboratory Animal Care protocols (#11799 and #13646) and the institutional guidelines set by the Veterinary Service Center at Stanford University.

## Additional files

### Supplementary file

• Supplementary file 1. Gene summaries.

### Major dataset

The following dataset was generated:

| Author(s) | Year | Dataset title | Dataset ID and/or URL | Database, license, and accessibility information |
|---|---|---|---|---|
| Smith CL, Spies N, Rodriguez JM, Baker JC, Batzoglou S, Sidow A | 2015 | Raw sequencing data | http://www.ncbi.nlm.nih.gov/geo/query/acc.cgi?acc=GSE62967 | Publicly available at NCBI Gene Expression Omnibus (GSE62967). |

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
