## [Decision Letter]

Thank you for sending your work entitled “Constraint and divergence of global gene expression in the mammalian embryo” for consideration at *eLife*. Your article has been evaluated by Chris Ponting (Senior editor), a Reviewing editor and three reviewers.

As listed below, the reviewers have communicated quite detailed major concerns about the manuscript. We would like to provide the opportunity for you to address each of the points raised and to provide an extensively revised manuscript before making a final decision. The Reviewing editor and the reviewers discussed their comments before we reached this decision, and the Reviewing editor has assembled the following comments to help you prepare a revised submission.

Major concerns:

1) The title is provocative, without obvious follow-up. ‘Convergence’ and ‘divergence’ has implications about genetic selection and systems properties. The reviewers expected a definition of what these concepts mean in this context, how the study design would address these questions, and then interpretation of how the results provide insight. This was not evident in this manuscript. The primary reference seems to be in the Introduction, where two definitions of divergence are provided: first, it refers to changes in embryonic development resulting from genotype-specific differences in maternal transcripts present in mature oocytes at fertilization, and second, divergence here seems to be differences in regulatory control between embryos and adults. Both have been extensively studied, first to understand the maternal:zygotic transition in gene expression patterns, and second to characterize differentiation of cells, organs, and tissues. Resolving these and other forms of direct (genetic) and indirect (epigenetic, social factors is beyond the scope of the present study design, except with weak ad hoc arguments.

2A) No mention about whether sex of embryos was determined, and results analyzed separately for males and females. Considerable literature shows that sex affects expression of many genes. If the authors have evidence that embryo sex is not important in this context, evidence and arguments need to be presented (see also comment 10 below).

B) Apparently RNAs from whole embryos were analyzed. But as the authors acknowledge, organogenesis is active at E11.5. So why study whole embryos rather than specific organs and tissues?

C) Paternal stress is an established factor affecting offspring phenotypes and molecular features. Typically these paternal factors are controlled by removing males from mating cages immediately after fertilization. Alternatively, in vitro fertilization is sometimes used. Regardless, these kinds of factors needed to be controlled in order to make strong conclusions about the nature of maternal effects.

3) The aspect of the paper that we found disappointing was the analysis of gene expression dynamics. The reviewers considered that the authors could get much more from this dataset. For example, they don't test for interactions between genotype and time/stage (somite number, this could be done quite easily even using PEER) and there is no consideration of non-motonic time trends etc. In short, we think the authors could likely find much more about how genetic variation influences expression dynamics by implementing something similar to the ideas of Francesconi (Nat 2014).

4) The next recommendation would be to try and get more from the comparison of ASE and eQTLs. The correlation is weak (R^2 =∼34% variance explained). What is the explanation for this? Biological or technical? Likely there is something interesting to say biological here about the causes of the differences between the hybrids and the recombinants.

5) In general, the results are not too surprising; for example, the observation (Results, third paragraph) that the expression of key developmental genes changes through development is to be expected, we are not sure this warrants a main text figure. Moreover, a significant number of eQTL have been found in other tissues/cell lines in mammals (multiple papers from the Dermitzakis and Pritchard labs), the majority of which have been shown to regulate gene expression in cis, so the novelty of this part of the manuscript is small.

6) Care must be taken when defining cis-regulatory variants identified from previous RNA-seq studies (Montgomery et al.; Pickrell et al.), as you have done in the Introduction. In particular, these studies focused on correlating variants near the gene of interest with its expression across individuals. In a number of cases, it was shown that the effect size of the eQTL was correlated with allele-specific expression levels, strongly suggesting that many variants act in cis. However, in most previous eQTL studies, variants that act in cis have typically been defined as variants that are proximal to the gene of interest. Some clarification of this subtle, but important distinction, would be useful here.

7) When you mention, in the Introduction, that, at E11.5, gastrulation has already occurred, and the embryo is already composed of multiple different cell types, how much of a concern is it that, across N2 embryos, the rate of development may differ (perhaps because of the genetic background), which might lead to heterogeneity in the composition of cell types profiled across embryos?

8) Results, second paragraph: Why was a paired t-test as opposed to a binomial test (perhaps with a beta prior to model over-dispersion) employed here? The latter seems a much more natural approach given the data.

9) Results, fourth paragraph: It would be interesting to study the overall expression levels of the genes considered here in a homogeneous genetic background (i.e., in the F0s). In particular, this would allow one to study how the expression of *Hbb-b1* and *Hbb-b2* differed during development—is it also the case that, in BL6, *Hbb-b1* increases in expression during development while *Hbb-b2* decreases? This would provide additional mechanistic insight and provide support for the compensatory mechanism alluded to in the last sentence.

10) Figure 3 and Results: What explains the strange patterns in Figure 3 where there seems to be a bimodal pattern of expression (for both alleles) for a subset of the genes (e.g., Zkscan5, Polm, Eif2ak and Rhob)? Is this related to differences in developmental stage and/or sex? To explore the latter, could the expression of genes on the X or Y chromosome be used to sex the embryos? More generally, this is a strange pattern, which provides some concern about the robustness of the selection of these genes as those that display strong maternal effects.

11) Figure 5 and Results: The comparison with previous studies is somewhat troubling. In particular, how were the data normalized to ensure the results were comparable (e.g., batch effects/control for potential confounding factors)? Also, a p-value cutoff, rather than an FDR cutoff, should be used to ensure results are comparable across studies (since the null distribution will differ). Without more information, it is difficult to assess the significance of this result.

12) Results: What functional categories were enriched for in the ASE genes? Also, for the enrichment analysis, did the background control for gene expression levels (i.e., were only expressed genes considered)?

---

## [Author Response]

*1) The title is provocative, without obvious follow-up. ‘Convergence’ and ‘divergence’ has implications about genetic selection and systems properties. The reviewers expected a definition of what these concepts mean in this context, how the study design would address these questions, and then interpretation of how the results provide insight. This was not evident in this manuscript. The primary reference seems to be in the Introduction, where two definitions of divergence are provided: first, it refers to changes in embryonic development resulting from genotype-specific differences in maternal transcripts present in mature oocytes at fertilization, and second, divergence here seems to be differences in regulatory control between embryos and adults. Both have been extensively studied, first to understand the maternal:zygotic transition in gene expression patterns, and second to characterize differentiation of cells, organs, and tissues. Resolving these and other forms of direct (genetic) and indirect (epigenetic, social factors is beyond the scope of the present study design, except with weak ad hoc arguments*.

We are so used to these terms in their evolutionary context that we did not ask ourselves whether it’s clear what we mean. We have changed the introductory paragraph to indicate that we use these terms in their standard evolutionary meaning and would be open to further clarification if necessary.

In evolutionary genetics, ‘constraint’ and ‘divergence’ are standard terms that describe the slowing of change, and the accumulation of change, respectively. Divergence is the evolutionary process that gives rise to differences; constraint is the brake purifying selection puts on divergence.

Specifically, in the context of our paper, the large number of genes with cis-eQTL and allele-specific expression, the change in the allelic ratios over developmental time (eg Hbb-b1 and Hbb-b2), and the maternal effect differences between the strains are indicative of gene regulatory divergence since the strains’ last common ancestor.

Similarly, ‘constraint’ refers to expression patterns that are more similar between strains than would be expected, indicating past purifying selection. We find evidence of constraint in the lack of trans-eQTL (fewer than would be expected, based on the comparison to adult mice), the higher sequence conservation and the depletion of regulatory gene ontology terms in the cis-eQTLs.

*2A) No mention about whether sex of embryos was determined, and results analyzed separately for males and females. Considerable literature shows that sex affects expression of many genes. If the authors have evidence that embryo sex is not important in this context, evidence and arguments need to be presented (see also comment 10 below)*.

This is an important point, and we thank the reviewers for mentioning it. We did indeed determine the sex of each embryo. We performed the suggested analysis and found only 4 additional genes with significant sex-specific expression patterns, even at higher (more lenient) FDR cutoffs. All these genes are on the X or Y chromosomes.

Sex differentiation begins with differential expression of Sry and then Sox9 in the genital ridges at E10.5. Our embryos were harvested at E11.5, before development of the gonads, which happens a day later at approximately E12.5. We do not expect substantial sex-specific differences in gene expression at E11.5 because most male-specific development begins with testosterone synthesis in the gonads. We also asked whether we can detect differences in genes that are known to be expressed differentially between the sexes in the mouse brain prior to gonad differentiation, but again only found a handful of X and Y chromosome genes to show any sign of sex bias.

Because the X and Y chromosomes are highly repetitive, and because we found only a few sex-biased genes, we considered only autosomal genes and markers for all subsequent analyses. A new paragraph has been added to the Methods to explain these analyses.

We note that we used sex as a covariate when performing batch normalization with PEER, thus controlling for any smaller sex-specific effects on gene expression.

*B) Apparently RNAs from whole embryos were analyzed. But as the authors acknowledge, organogenesis is active at E11.5. So why study whole embryos rather than specific organs and tissues*?

An E11.5 embryo is, on average, approximately 5x4x3mm^3^. Attempting to dissect individual organs, which are very small and morphologically not particularly distinct at this stage of development, would introduce substantial confounding factors. The most important and the most difficult one to control is variation among individual dissections, such as in the precise cut separating the head from the body. This introduces unknown and uncontrollable variance that potentially results in apparent differences in gene expression. Also, a ‘tissue’ isolated in any practically feasible dissection, for example head vs. body vs. limbs, is still a heterogeneous mix of developing cells. Finally, dissection and subsequent sequencing of individual organs from hundreds of mice would have multiplied costs to the extent that it would have been prohibitively expensive.

We obtained definitive results with important conclusions despite the perceived shortcoming of using whole embryos. Future studies on individual embryonic tissues, if executed properly, may refine our findings.

*C) Paternal stress is an established factor affecting offspring phenotypes and molecular features. Typically these paternal factors are controlled by removing males from mating cages immediately after fertilization. Alternatively, in vitro fertilization is sometimes used. Regardless, these kinds of factors needed to be controlled in order to make strong conclusions about the nature of maternal effects*.

We used standard mating procedures. One male was placed into a cage with 2 females in the evening and removed promptly on the morning after mating (E0.5), minimizing any effect of paternal behavior on the mother’s pregnancy. We have added a note to this effect in the Methods.

*3) The aspect of the paper that we found disappointing was the analysis of gene expression dynamics. The reviewers considered that the authors could get much more from this dataset. For example, they don't test for interactions between genotype and time/stage (somite number, this could be done quite easily even using PEER) and there is no consideration of non-motonic time trends etc. In short, we think the authors could likely find much more about how genetic variation influences expression dynamics by implementing something similar to the ideas of Francesconi (Nat 2014)*.

We thank the reviewers for pressing this point, which we had extensively explored during preparation of the manuscript. In fact, we were quite excited to find 12 genes with significant genotype x time interactions, which we found by correlating ASE and somite number. Thus, for these 12 genes, cis-regulatory polymorphisms cause the B6 and Cast alleles to be expressed differently over time. We highlighted in the text an interesting subset of these heterochronic genes, the beta-globins (also see point 9 below).

The time course we assayed is relatively short, ranging from approximately E11.0 to E12.0, or about 5% of total pre-term developmental time. We therefore expect only a very small fraction of genes to show non-monotonic trends in expression, and even fewer where that trend is polymorphic between the assayed strains. Thus, we do not have the same expectation as the reviewers regarding potential widespread, genetically controlled heterochrony.

In order to definitively explore this issue, we applied the Francesconi (2014) method to explicitly identify genotype x time interactions. This method uses an F-test to assess whether a spline-based genotype x time model (with interaction) fit the data better than a genotype + time model (additive effects only). We found some interactions, but all genome-wide significant results were the result of overfitting, a pitfall that is particularly prevalent with non-monotonic trends. In order to avoid overfitting, we modified the splines approach to use resampling (rather than an F-test) to identify significant genes. This approach yielded no significant genes. We also asked whether genotype x time interactions could be observed using simple linear models only, but again were unable to find any convincing significant results. This underscores the greater power of our straightforward correlation approach, which did yield 12 heterochronic genes.

We considered the reviewers’ suggestion to use PEER to test for interactions between genotype and time/stage. Typically, PEER is used to remove batch effects, including both known and unknown covariates. Since somite number is a known covariate in our experiment, it was unclear to us how PEER would be useful in identifying time x genotype interactions, since we can model time explicitly.

Based on these results, we conclude that the divergence between Cast and B6 produced a small number of significant time x genotype interactions that are active at E11.5. That we found any is a testament to the robustness of our data and a reflection of biological reality in the developing mammalian embryo. Future studies may refine or expand these results.

*4) The next recommendation would be to try and get more from the comparison of ASE and eQTLs. The correlation is weak (R^2 =∼34% variance explained). What is the explanation for this? Biological or technical? Likely there is something interesting to say biological here about the causes of the differences between the hybrids and the recombinants*.

The explanation is technical. Most genes do not exhibit ASE or eQTL effects, and therefore contribute noise around (0.5, 0.5). When considering only the genes with significant ASE or eQTL, agreement improves substantially and we find that only about 1% of genes are discordant between ASE and eQTL. The correlation improves from rho=0.58 when considering the top 5000 genes with best ASE read coverage to rho=0.70 when considering the top 1000 genes.

In comparison to previous studies the correlation we find between eQTL and ASE appears to be quite strong. For example, Lagarrigue et al. (2013) compared ASE and eQTL in mice, finding 90% of significant eQTL genes showed allelic ratios in the same direction as predicted by eQTL, but the overall quantitative correlation appeared to be poor. Similarly, Battle et al. (2014) and Li et al. (2014) show highly significant correlations between ASE and eQTL effect sizes, but the Pearson correlations are at most 0.68 when considering genes with deep allele-specific read coverage.

*5) In general, the results are not too surprising; for example, the observation (Results, third paragraph) that the expression of key developmental genes changes through development is to be expected, we are not sure this warrants a main text figure*.

The differential developmental regulation of the globins between B6 and Cast highlights fundamental physiological changes due to divergence in gene expression regulation between the strains. That the two adult globins show opposite patterns is suggestive of compensatory evolutionary dynamics within this gene family. We are not aware of any such finding in mammalian embryogenesis and while one might have expected such phenomena, our study is the first to show that they actually occur.

Figure 2 establishes that our data recapitulate the previously reported shift from fetal to adult hemoglobins. This is important context for Figure 2, which demonstrate that the two strains initiate this transition using different globin family members.

Our study is also the first to show widespread, substantial effects of maternal genotype on embryonic expression. Is that ‘surprising’? We think it is, and it’s important too.

*Moreover, a significant number of eQTL have been found in other tissues/cell lines in mammals (multiple papers from the Dermitzakis and Pritchard labs), the majority of which have been shown to regulate gene expression in cis, so the novelty of this part of the manuscript is small*.

The novelty here is not the abundance of cis-eQTL; it is the paucity of trans-eQTL. Previous studies in model organisms, including adult mice, have typically shown that 10–20% of eQTL are in trans (e.g. Chow et al. 2015), with the rest acting locally and probably therefore in cis. We demonstrate that there is a highly significant quantitative difference between adults and embryos, with only approximately 1% of all eQTL acting in trans in embryos. Together with the sequence conservation analysis of eQTL/ASE genes, our results show that there is stronger evolutionary constraint on mammalian gene regulatory networks in embryos than in adults. This is a novel, non-obvious, conclusion.

*6) Care must be taken when defining cis-regulatory variants identified from previous RNA-seq studies (Montgomery et al.; Pickrell et al.), as you have done in the Introduction. In particular, these studies focused on correlating variants near the gene of interest with its expression across individuals. In a number of cases, it was shown that the effect size of the eQTL was correlated with allele-specific expression levels, strongly suggesting that many variants act in cis. However, in most previous eQTL studies, variants that act in cis have typically been defined as variants that are proximal to the gene of interest. Some clarification of this subtle, but important distinction, would be useful here*.

We agree, and thank the reviewers for pointing that out. The relevant sections have been edited to make clear that genomic proximity does not necessarily imply cis-regulation.

*7) When you mention, in the Introduction, that, at E11.5, gastrulation has already occurred, and the embryo is already composed of multiple different cell types, how much of a concern is it that, across N2 embryos, the rate of development may differ (perhaps because of the genetic background), which might lead to heterogeneity in the composition of cell types profiled across embryos*?

We think that variance in developmental rate would simply dilute our power to detect effects. The developmental ASE effects we uncover (Figure 2) are inconsistent with high variance in developmental progress, at least in the tissues in which these genes are expressed.

*8) Results, second paragraph: Why was a paired t-test as opposed to a binomial test (perhaps with a beta prior to model over-dispersion) employed here? The latter seems a much more natural approach given the data*.

We carefully considered several statistical approaches before settling on the paired t-test, which has two key appealing features:

A) It considers the mean and variance across many individuals; since we considered only heterozygous sites in N2s, this number varies by individual but is on average ∼75.

B) It takes advantage of the within-individual comparison (i.e., for each individual, the B6 allele read count is subtracted by the Cast allele read count); this reduces the effects of biological and technical noise between samples because much of this noise should affect both alleles similarly.

Previous studies have used the beta-binomial distribution to model the count data underlying ASE measurements (e.g. Skelly et al. 2011). Such models are unlikely to provide improvements to our results while adding substantial complexity to the analysis as well as free parameters whose data-based estimation may result in overfitting. Importantly, we found that p-values obtained from the t-test correlated well with those calculated by resampling, but were slightly conservative.

We have expanded our note in the Methods explaining our decision to use the paired t-test.

*9) Results, fourth paragraph: It would be interesting to study the overall expression levels of the genes considered here in a homogeneous genetic background (i.e., in the F0s). In particular, this would allow one to study how the expression of Hbb-b1 and Hbb-b2 differed during development—is it also the case that, in BL6, Hbb-b1 increases in expression during development while Hbb-b2 decreases? This would provide additional mechanistic insight and provide support for the compensatory mechanism alluded to in the last sentence*.

Both *Hbb-b1* and *Hbb-b2*, the adult beta-globins, are increasing in total expression during the time course we assayed (around E11.5), while the embryonic globins are slowly being downregulated. Our ASE data, which are particularly well-suited for such analyses because both alleles are assayed in the same embryo, demonstrate that the Cast allele of *Hbb-b1* becomes activated earlier than the B6 allele, and that the Cast allele of *Hbb-b2* becomes activated later than the B6 allele. In support of the ASE data, the regression slopes for total expression (a coarser metric than ASE) differ between heterozygotes and homozygotes for both genes, significantly so for *Hbb-b2* (p=0.04).

Therefore, there are one or more cis-regulatory differences between Cast and B6 in the Hbb locus, which we detect across a large sample of N2 embryos. Only a violation of basic rules of inheritance would generate a different result in the F0s.

*10) Figure 3 and Results: What explains the strange patterns in Figure 3 where there seems to be a bimodal pattern of expression (for both alleles) for a subset of the genes (e.g., Zkscan5, Polm, Eif2ak and Rhob)? Is this related to differences in developmental stage and/or sex? To explore the latter, could the expression of genes on the X or Y chromosome be used to sex the embryos? More generally, this is a strange pattern, which provides some concern about the robustness of the selection of these genes as those that display strong maternal effects*.

We found that the genes with a bimodal pattern of expression were correlated in their expression patterns across embryos, suggestive of a concerted regulatory program. We know each embryo’s sex (as explained above) and found no evidence that these expression patterns were caused by sex of the embryo or developmental timing. Instead, we found these genes loosely clustered on the basis of litter, with most outliers comprised of embryos from a small number of litters. This clustering by litter suggests that there may be influences beyond genotype of the mother that result in maternal effects. This result does not impact our conclusions about the effects of maternal genotype on embryonic expression.

*11) Figure 5 and Results: The comparison with previous studies is somewhat troubling. In particular, how were the data normalized to ensure the results were comparable (e.g., batch effects/control for potential confounding factors)? Also, a p-value cutoff, rather than an FDR cutoff, should be used to ensure results are comparable across studies (since the null distribution will differ). Without more information, it is difficult to assess the significance of this result*.

We completely agree on the importance of making sure that results are comparable between the studies and have now gone to considerable additional lengths to do so. In response to the comment, we have batch normalized the previously published eQTL data using PEER, and have indicated this in the Methods section. In addition, we have performed two new analyses comparing the rates of trans-eQTL in our embryo study to the previous ones of adult mice.

First, as the reviewers suggest, we have compared the proportion of trans-eQTL at differing p-value cutoffs. Second, we have compared the proportion of trans-eQTL as the total numbers of eQTL increase. See Figure 6. Both of these alternative analyses support the original conclusion that adult mouse tissues show a substantially higher proportion of trans-eQTL than do embryos. Furthermore, the results hold when sampling smaller or larger subsets of the previously published datasets.

Author response image 1.Fraction of eQTL in trans, comparing current embryo study to previous adult studies. (**A**) eQTL were calculated at increasing p-value cutoffs (x-axis). (**B**) eQTL were compared at increasing ranks (e.g., top 500 eQTL from each study). See also Figure 5.**DOI:**
http://dx.doi.org/10.7554/eLife.05538.015

As the reviewers point out, the null distributions will be different between studies, due to differences in the number of correlated markers, differences in which genes are expressed, etc. Thus, we believe that the raw p-values should not be directly compared. In contrast, for each dataset, the false discovery rate is calculated relative to its permuted distribution, thereby controlling for differences in the null distribution between studies. Therefore, we are keeping the main figure using FDR, but have added a note in the text about the robustness of the result to variations of the analysis.

*12) Results: What functional categories were enriched for in the ASE genes? Also, for the enrichment analysis, did the background control for gene expression levels (i.e., were only expressed genes considered)*?

The top enriched categories for ASE genes were “cellular biosynthetic process” and “ribonucleoprotein complex”. We did not mention these results in the text, as we were unsure about how to interpret them biologically and, based on our other results, we did not expect substantial positive selection.

All Gene Ontology analyses were performed using background sets containing only expressed genes (and the ASE analyses using only genes with read coverage of SNPs, informative for the ASE analyses). We have noted this in the Methods.